# Raspberry Ketone [4-(4-Hydroxyphenyl)-2-Butanone] Differentially Effects Meal Patterns and Cardiovascular Parameters in Mice

**DOI:** 10.3390/nu12061754

**Published:** 2020-06-11

**Authors:** Dushyant Kshatriya, Lihong Hao, Xinyi Li, Nicholas T. Bello

**Affiliations:** 1Nutritional Sciences Graduate Program, School of Environmental and Biological Sciences, Rutgers, The State University of New Jersey, New Brunswick, NJ 08901, USA; dsk118@scarletmail.rutgers.edu (D.K.); cindyx.li@rutgers.edu (X.L.); 2Department of Animal Sciences, School of Environmental and Biological Sciences, Rutgers, The State University of New Jersey, New Brunswick, NJ 08901, USA; haoli@sebs.rutgers.edu

**Keywords:** frambinone, meal frequency, open-field test, elevated plus maze, sensory motor gating, pre-pulse inhibition, c-Fos

## Abstract

Raspberry ketone (RK; [4-(4-hydroxyphenyl)-2-butanone]) is a popular nutraceutical used for weight management and appetite control. We sought to determine the physiological benefits of RK on the meal patterns and cardiovascular changes associated with an obesogenic diet. In addition, we explored whether the physiological benefits of RK promoted anxiety-related behaviors. Male and female C57BL/6J mice were administered a daily oral gavage of RK 200 mg/kg, RK 400 mg/kg, or vehicle for 14 days. Commencing with dosing, mice were placed on a high-fat diet (45% fat) or low-fat diet (10% fat). Our results indicated that RK 200 mg/kg had a differential influence on meal patterns in males and females. In contrast, RK 400 mg/kg reduced body weight gain, open-field total distance travelled, hemodynamic measures (i.e., reduced systolic blood pressure (BP), diastolic BP and mean BP), and increased nocturnal satiety ratios in males and females. In addition, RK 400 mg/kg increased neural activation in the nucleus of the solitary tract, compared with vehicle. RK actions were not influenced by diet, nor resulted in an anxiety-like phenotype. Our findings suggest that RK has dose-differential feeding and cardiovascular actions, which needs consideration as it is used as a nutraceutical for weight control for obesity.

## 1. Introduction

Raspberry ketone (RK; 4-(4-hydroxyphenyl)-2-butanone) is a naturally occurring phenolic compound responsible for the aroma and flavor of raspberries (*Rubus idaeus*) [1,2]. Naturally derived RK is costly to produce, therefore synthetic sources of RK are readily available [3,4]. In the United States, RK is designated as a generally recognized as safe (GRAS) food additive and is listed by the Food and Drug Administration (FDA) as a synthetic flavoring substance [5]. Recently, RK has been marketed as a weight loss agent and an appetite suppressant, and used as single- or multi-ingredient supplement [6,7,8,9].

A few studies reveal the efficacy of RK in the prevention of fat accumulation. RK has been shown to reduce lipid accumulation and alter expression of lipolytic and adipogenic genes in 3T3-L1 adipocytes [10,11,12,13,14]. Further, it can increase fat oxidation in vitro and the effect may be mediated by heme oxygenase-1 and brown-like adipocyte formation [13,14]. RK mitigated ovariectomy-induced weight gain [12,14], and reduced high-fat diet induced nonalcoholic steatohepatitis in rats [15]. Adulteration of diet with RK has shown to prevent weight gain [16,17], however, the strong sensory profile of RK could have potentially affected food intake. Previous work from our lab supports the preventative actions of oral gavage administration of RK against weight gain and fat accumulation in a high-fat diet induced obesogenic environment [18]. Along with management of weight gain, RK can serve cardio protective roles. RK pretreatment reduced isoproterenol-induced cardiac tissue damage and dyslipidemia [19,20]. Because obesity therapeutic agents often are associated with adverse cardiovascular events [21], the RK doses that prevent weight gain should also be examined for their hemodynamic effects. Indeed, weight loss nutraceuticals are typically consumed on a long-term basis for several months, therefore, understanding the cardiovascular risks and benefits are critical for their assessment.

Diet-induced obesity often involves hyperphagia and changes in feeding patterns such as increased meal frequency and meal size [22,23]. Therefore, it is valuable to understand the influence of RK, as a preventative weight loss agent, on meal patterns. The present study was designed to characterize the dose-dependent effect of RK on meal microstructure in male and female mice. In addition, we assessed the interaction of diet and RK dose on hemodynamic parameters. We hypothesized that there would be a dose-dependent preventative effect of RK on the high-fat diet- induced alterations in meal patterns and cardiovascular outcomes. Because weight changes have been associated with anxiogenic or antipsychotic agents [24,25], we also conducted a series of behavioral tests to determine whether the effects were secondary to an anxiogenic effect of RK or impairments in sensory-motor gating.

Previous bioavailability studies from our laboratory have shown that oral RK is rapidly absorbed and a 200 mg/kg dose can peak in the plasma after 15 min in male and female mice [26]. In addition, RK and metabolites were detected in the brain and white adipose tissue of normal weight and diet-induced obese mice [27]. While there is accumulating evidence that RK has direct actions in adipose tissue [10,11,13], the ability of RK to activate brain areas involved in feeding and cardiovascular control has not been explored. One central site that overlaps to regulate these two physiological systems, is the nucleus of the solitary tract (NTS) in the caudal hindbrain [28,29,30]. Therefore, we examined whether RK, at doses that influence meal patterns and hemodynamic parameters, also activates the NTS and associated area postrema (AP).

## 2. Materials and Methods

### 2.1. Mice

A total of 270 male and female C57BL/6J mice (7 weeks old) were purchased from The Jackson Laboratory (Bar Harbor, ME, USA). Mice were fed ad libitum standard chow (Purina Mouse Diet 5015, St. Louis, MO, USA; 25.34% fat, 19.81% protein, 54.86% carbohydrate, 3.7 kcal/g) upon arrival. One week later, mice (8 weeks old) were equally divided by body weight and were switched to a high-fat diet (HFD; D12451, Research Diets, Inc., New Brunswick, NJ; 45% fat, 20% protein, 35% carbohydrate; 4.73 kcal/g) or sucrose-matched control diet (LFD; D12450H; Research Diets, Inc., New Brunswick, NJ; 10% fat, 20% protein, 70% carbohydrate; 3.85 kcal/g) and fed ad libitum. Water was available at all times. Mice were single-housed, unless otherwise noted, and maintained on a 12-h light and 12-h dark cycle with lights on from 07:00 h to 19:00 h. The animal care protocol was approved by the Institutional animal Care and Use Committee of Rutgers University (OLAW #A3262-01). Coincident with the diet switch, daily oral dosing was initiated in mice with the following treatments: vehicle (VEH; 50% propylene glycol, 40% water, and 10% dimethyl sulfoxide (DMSO) or raspberry ketone (RK, 200 mg/kg or 400 mg/kg; (4[4-hydroxyphenyl]-2-butanone; 99%; cat#178519; Sigma Aldrich). A reference sample from each lot number of raspberry ketone batch was deposited in a secure, climate-controlled repository [31]. Oral dosing of the mice was performed using single-use, sterile plastic feeding tubes (20 ga × 30 mm; cat# FTP-20-30, Instech Laboratories, Plymouth Meeting, PA, USA). Mice were fed the individual diets and oral dosed for 2 weeks, unless otherwise noted. Daily body weight and cumulative food intake were measured throughout the entire dosing period. Daily dosing was performed between 10:00h and 12:00h. Mice were divided into six groups based on diet and dose; HFD-Vehicle, LFD-Vehicle, HFD-RK (200 mg/kg), LFD-RK (200 mg/kg), HFD-RK (400 mg/kg), and LFD-RK (400 mg/kg), see Appendix A
Appendix A.

### 2.2. Meal Patterns and Meal Microstructures

Meal microstructures were analyzed in male and female C57BL/6J mice (n = 4 – 16 per group/sex) using the Biological Data Acquisition System (BioDAQ; Research Diets, New Brunswick, NJ, USA). This system utilizes standard shoe-box style cages with a gated front-mounted food hopper. The gated hopper sits upon a sensor that detects net changes in food weight per second. A feeding bout was defined as a change in stable weight of the hopper. Bouts were clustered into meals, defined by an inter-meal interval of 300 s and a minimum of 0.02 g consumed. Meal patterns were measured over the entire two-week period. Data were analyzed and averaged over the 14-day period and presented separately for the dark-cycle (nocturnal) and light-cycle (diurnal). Data recordings in the BioDAQ computer were paused when the mice were handled for measurement of body weight and dosing. Daily data were used to calculate, meal frequency (meals/day), meal size (kcal), meal duration (min/meal), eating rate (kcal/min), and satiety ratio (Inter meal interval (min)/kcal). Mice that demonstrated food shredding were not included in the analyses [18].

### 2.3. Anxiety-Related and Sensorimotor Gating Behavioral Testing

After the BioDAQ data collection, mice were kept in their respective diet and daily dosing groups for an additional 8 days and exposed to three behavioral tests. Mice were exposed to the following tests, one teste per day in the following order: open field test (Days 18–20), elevated plus maze (Days 19–21), and pre-pulse inhibition (Days 20–22). Daily dosing and diet regimen continued until completion of all behavioral tests. Daily testing times were staggered and were performed >1 h after mice received their respective daily dose.

#### 2.3.1. Open Field

The open field tests took place in a brightly lit box (40 × 40 × 38 cm) with white floors and luminescent walls. Mice were placed in the center of the apparatus for a ten-minute test period. Measurements of locomotion, exploration, and anxiety were video recorded and scored offline by an individual blinded to the grouping of the study [32].

#### 2.3.2. Elevated Plus Maze

Mice were exposed to the elevated plus maze apparatus for 5 min. The elevated plus maze apparatus has two open arms (25 × 5 × 0.5 cm) across from each other and perpendicular to two closed arms (25 × 5 × 16 cm) with a center platform (5 × 5 × 0.5 cm) and is 50 cm above the floor. At the commencement of the test, each mouse was placed at the junction of the open and closed arms and faced the open arm. The number of entries into open and closed arms, risk assessment, number of feces, and self-grooming behavior were video recorded for five minutes and scored offline by an individual blinded to the study [33].

#### 2.3.3. Pre-Pulse Inhibition

Each pre-pulse inhibition session was conducted in a ventilated soundproof startle chamber (Med Associates Inc., Fairfax, VT, USA) and proceeds with a 5-min acclimation period with 70 dB background noise followed by five successive 110 dB trials for habituation. Six different trial types were then presented: startle pulse (ST110, 110 dB/40 ms), low prepulse stimulus given alone (P74, 74 dB/20 ms), high prepulse stimulus given alone (P90, 90 dB/20 ms), P74 or P90 given 100 ms before the onset of the startle pulse (PP74 and PP90, respectively), and finally a trial where only the background noise was presented in order to measure the baseline movement in the cylinders. All trials were applied ten times and presented in random order (P74 and P90 were only given five times) and the average inter-trial interval was 15 s (10–20 s) [34].

### 2.4. Hemodynamic Measures

In a separate set of male and female C57BL/6J mice (n = 11–12 per group/sex) measurements of hemodynamic parameters were performed using the noninvasive blood pressure CODA system (Kent Scientific, Torrington, CT, USA) system. This computerized system measures systolic and diastolic blood pressure, mean blood pressure, and heart rate via tail volume pressure recordings. Animals were acclimated to the holder and cuff of the CODA system for 5 min before each recording trial. CODA trials were performed 1–2 h after daily dosing of either RK or vehicle. Animals were exposed to 5 consecutive days of recording trials, data from the last 2 days were used for data analysis (i.e., days 13 and 14 of the 2-week dosing period), see Appendix A
Appendix A.

### 2.5. Immunohistochemistry of Area Postrema (AP) and Nucleus of The Solitary Tract (NTS)

A separate set of male C57BL/6J mice (n = 5–6 per group) were dosed with RK (400 mg/kg) or vehicle for 14 days. On day 14, all mice were returned to their cage without food or water and left undisturbed for 120 min. Mice were then deeply anesthetized with 0.1% euthasol (pentobarbital sodium and phenytoin sodium) solution intraperitoneal (IP), exsanguinated with 0.9% saline, and perfused with 4% paraformaldehyde in phosphate buffered saline (PBS). Brains were extracted and post-fixed for 24 h in 4% paraformaldehyde in PBS, then switched to 20% sucrose in 4% paraformaldehyde until sectioning. Free-floating sections (40 μm) of the forebrain were obtained by using a Leica cryostat (Leica Microsystems, Rijswijk, The Netherlands). Sections were stored in cryoprotectant until immunohistochemistry was performed. Sections were transferred to a new clean plate containing PBS (10 mM phosphate, 150 mM NaCl, pH 7.5). Initial PBS was removed, sections were then washed 3 × 10 min in PBS. Endogenous peroxidases were neutralized with 0.3% H_2_O_2_ in dH_2_O. After a 3 × 10 min PBS wash, sections were incubated in normal goat serum (PK-4001, Vectastain ABC kit, Vector Laboratories, Burlingame, CA, USA) with 0.3% Triton-X-100 in PBS for 30 min. c-Fos immunolabeling was performed with a polyclonal rabbit IgG anti c-Fos antibody (ab190289, Abcam, Cambridge, MA, USA), diluted 1:100 in PBS. Tissue sections were incubated overnight (~20 h). Sections were transferred to a new clean plate, washed 3 × 10 min in 0.1% Triton X-100 in PBS, then incubated for 30 min in biotinylated secondary antibody (goat IgG anti-rabbit, PK-4001, Vectastain ABC kit, Vector Laboratories) with 0.3% Triton X-100 in PBS. After 3 × 10 min wash in PBS, sections were incubated in an avidin-peroxidase complex (PK-4001, Vectastain ABC kit, Vector Laboratories) for 45 min. Sections were washed 3 × 10 min in PBS. Staining was performed using nickel diaminobenzidine tetrahydrochloride (Ni-DAB) Chromagen (SK-4100, DAB Peroxidase Substrate Kit, 3,3′-diaminobenzidine, Vector Laboratories) for approximately 30 s to stain Fos-like products black. PBS was added immediately after desired stain was reached and sections were washed in 3 × 10 min in PBS to halt the Ni-DAB reaction. Sections were mounted on gelatin-coated slides (Fisherbrand Double Frosted Microscope Slides, Thermo Fisher Scientific Inc., Bridgewater, NJ, USA) and dehydrated with ethanol and xylenes prior to coverslipping with permount [35], see Appendix A
Appendix A.

### 2.6. Imaging and Quantification of C-Fos Positive Nuclei

Coronal sections from area postrema (AP) and four rostrocaudal levels of the nucleus of the solitary tract (NTS) were analyzed per animal. The anterior-posterior levels were determined by coordinates from Bregma [36]. The NTS areas consisted of anatomically matched sections from caudal (cNTS; −7.92 mm), at the level of the obex, corresponding to the posterior edge of the AP; medial (mNTS; −7.48 mm) at the maximal extent of the AP; intermediate (iNTS; −7.08 mm), anterior to the AP, corresponding to the maximal extent of the gelatinous subnucleus of the NTS; rostral (rNTS; −6.84 mm) consisting of the area rostral to the gelatinous nucleus and the caudal aspect of the medial vestibular nucleus on the dorsal boundary [36]. This analysis provided a view for the rostral-caudal extent of c-Fos activation [29]. Imaging was performed using an Olympus FSX-BSW imaging scope and FSX100 software (Olympus videoscope, Tokyo, Japan). Quantification was performed by identifying c-Fos positive black nuclei using Image J software system (NIH, Bethesda, MD, USA) image analysis software. Three anatomically matched tissue slices of each region (unilateral) of each mouse were used in data analysis. Cells were counted by two observers blinded to the experimental conditions [35].

### 2.7. Statistical Analyses

Data are presented as means ± standard error of the mean (SEM). Separate two-way analysis of variance (ANOVA) or two-way ANOVA with repeated measures were performed to determine the effects of treatment conditions on individual measured parameters. When justified, Newman-Keuls post-hoc tests were performed unless otherwise specified. All statistical and power analyses were performed using Statistica 7.1 software (StatSoft; Dell Inc, Round Rock, TX, USA) and significance was set at α = 0.05.

## 3. Results

### 3.1. Bodyweights Over 14-Day Dosing and Diet Acesss

At the start of the experiments and group assignment, males baseline body weights were 23.2 ± 0.3 g for HFD-Vehicle, 23.3 ± 0.5 g for HFD-RK (200 mg/kg), 23.3 ± 0.5 g for HFD-RK (400 mg/kg), 23.1 ± 0.4 g for LFD-Vehicle, 23.6 ± 0.4 g for LFD-RK (200 mg/kg), and 23.1 ± 0.6 g for LFD-RK (400 mg/kg). For body weight gain over the 14-day diet and dosing regimen, there were effects for diet (F (1, 58) = 9.6, *p* < 0.005) and dose (F (2, 58) = 6.9, *p* < 0.005), days (F (12, 696) = 37.9, *p* < 0.005) and dose x days (F (24, 696) = 2.2, *p* < 0.001). There was increased body weight gain in the HFD fed male mice (*p* < 0.05). The 400 mg/kg dose produced an overall reduction in body weight gain over the 14-days for males (*p* < 0.05) compared with vehicle dose. There were body weight reductions with 400 mg/kg, compared with vehicle, for days 2–5 of dosing (*p* < 0.05 for all days), see Figure 1A. For females, baseline body weights were 18.3 ± 0.5 g for HFD-Vehicle, 18.1 ± 0.3 g for HFD-RK (200 mg/kg), 18.2 ± 0.4 g for HFD-RK (400 mg/kg), 18.4 ± 0.4 g for LFD-Vehicle, 17.8 ± 0.5 g for LFD-RK (200 mg/kg), and 17.9 ± 0.4 g for LFD-RK (400 mg/kg). For body weight gain there were effects for diet (F (1, 41) = 4.2, *p* < 0.05), dose (F (2, 41) = 3.2, *p* < 0.05), and days (F (12, 492) = 43.0, *p* < 0.0005). There was an increase in body weight gain in the HFD fed female mice (*p* < 0.05). The 400 mg/kg dose produced a reduction in body weight gain, compared with 200 mg/kg, over the 14-days for females (*p* < 0.05), see Figure 1B.

### 3.2. Meal Pattern Analysis Over 14 Days Dosing and Diet Access

For nocturnal meal frequency, in males there were diet (F (1, 54) = 22.9, *p* < 0.005) and dose (F (1, 54) = 13.5, *p* < 0.005) effects. More meals were consumed in male mice with HFD than LFD (*p* < 0.001). Fewer meals were consumed by RK 200 mg/kg (*p* < 0.001) and RK 400 mg/kg (*p* < 0.05) dosed mice, compared with vehicle. In females, there were diet (F (1, 34) = 46.9, *p* < 0.001) and dose (F (2, 34) = 11.9, *p* < 0.05) effects. More meals were consumed in female mice with HFD than LFD (*p* < 0.05). More meals were consumed with RK 200 mg/kg compared with RK 400 mg/kg (*p* < 0.001) and vehicle (*p* < 0.05), see Figure 2A. For nocturnal meal size, in males there was a dose (F (2, 54) = 7.6, *p* < 0.05) effect. Meal sizes were increased by RK 200 mg/kg (*p* < 0.05). In females, there were diet (F (1, 34) = 9.6, *p* < 0.001) and dose (F (2, 34) = 4.3, *p* < 0.05) effects. Meal sizes were increased by LFD (*p* < 0.05) and decreased by RK 200 mg/kg compared with RK 400 mg/kg and vehicle (*p* < 0.05 for both). For nocturnal meal duration, in males there were diet (F (1, 54) = 56.5, *p* < 0.001) and dose (F (2, 54) = 11.3, *p* < 0.001) effects. Meal duration was shorter with HFD compared with LFD (*p* < 0.005) and RK 200 mg/kg dose increased meal duration (*p* < 0.001). In females, there was a diet effect (F (1, 34) = 30.7, *p* < 0.005) for meal duration. Meal duration was shorter with HFD (*p* < 0.05), see Figure 2C. For nocturnal eating rate, in males there were diet (F (1, 54) = 18.9, *p* < 0.001) and dose (F (2, 54) = 4.9, *p* < 0.05) effects. There was an increase in eating rate with HFD (*p* < 0.001), and RK 400 mg/kg had a higher eating rate compared with VEH and RK 200 mg/kg (*p* < 0.05 for both). In females, there were diet (F (1, 32) = 43.6, *p* < 0.005) and diet X dose (F (2, 32) = 4.4, *p* < 0.05) effect for eating rate. In HFD, there was increase in eating rate compared with LFD (*p* < 0.005) and 200 mg/kg had the highest eating rate of all groups (*p* < 0.05), see Figure 2D. For nocturnal satiety ratio, in males, there were diet (F (1, 54) = 5.2, *p* < 0.05) and dose (F (2, 54) = 6.2, *p* < 0.005) effects. Satiety ratio was higher in male mice fed HFD than LFD (*p* < 0.05). RK 200 mg/kg and 400 mg/kg increased satiety ratio compared with vehicle (*p* < 0.05 for both). For satiety ratio in females, there was a dose effect (F (2, 32) = 5.9, *p* < 0.05), with higher satiety ratio in female mice receiving RK 400 mg/kg (*p* < 0.05), see Figure 2E.

There were no effects of treatment on diurnal meal size, meal duration, and satiety ratio, in male mice. For meal frequency, there was diet X dose (F (2, 54) = 3.5, *p* < 0.05) effect, with an increased meal number in male mice receiving HFD and RK 200 mg/kg. There was a diet effect on eating rate (F (1, 54) = 11.6, *p* < 0.05), with a higher rate in HFD fed mice. There were no effects of treatment on diurnal meal frequency, satiety ratio and eating rate in female mice. There was an effect of dose (F (2, 34) = 6.9, *p* < 0.05) on diurnal meal size, with an increase in RK 400 mg/kg compared with RK 200 mg/kg and vehicle (*p* < 0.05 for both). For diurnal meal duration in female, there was an effect of diet (F (1, 34) = 18.7, *p* < 0.005), with shorter meals in HFD compared with LFD fed mice.

At the end of the 14-day meal pattern analysis the male cumulative food intake was 147.3 ± 2.8 kcal for HFD-Vehicle, 137.3 ± 2.2 kcal for HFD-RK (200 mg/kg), 136.2 ± 4.3 kcal for HFD-RK (400 mg/kg), 130.9 ± 3.2 kcal for LFD-Vehicle, 135.5 ± 2.3 kcal for LFD-RK (200 mg/kg), and 132.1 ± 13.6 kcal for LFD-RK (400 mg/kg). The female cumulative food intake was 177.5 ± 12.3 kcal for HFD-Vehicle, 175.3 ± 4.7 kcal for HFD-RK (200 mg/kg), 156.1 ± 7.1 kcal for HFD-RK (400 mg/kg), 152.2 ± 9.4 kcal for LFD-Vehicle, 149.5 ± 3.1 kcal for LFD-RK (200 mg/kg), and 133.1 ± 6.6 kcal for LFD-RK (400 mg/kg). There was no effect of treatment on cumulative food intake.

### 3.3. Open-Field After 14 Days of Dosing and Diet Access

For time spent in center, in males there was an effect of dose (F (2, 55) = 5.6, *p* < 0.01). Male RK 200 mg/kg mice spent more time in the center of the open field than VEH (*p* < 0.05) and RK 400 mg/kg (*p* < 0.01), see Figure 3A. There was a significant effect of dose (F (2, 55) = 5.8, *p* < 0.01) on the number of entries into the outer zone in male mice, with RK 400 mg/kg mice entering the outer zone fewer times than VEH (*p* < 0.01). Similarly, dose (F (2, 41) = 5.2, *p* < 0.05) had a significant effect on number of entries of female mice into outer zone, with RK 400 mg/kg mice entering the outer zone fewer times than VEH (*p* < 0.05) and RK 200 mg/kg (*p* < 0.05) mice, see Figure 3B. Whereas, in male mice both dose (F (2, 55) = 6.7, *p* < 0.01) and diet (F (1, 55) = 10.0, *p* < 0.01) had an effect on total distance travelled. RK 400 mg/kg treated male mice travelled less than VEH (*p* < 0.01), and HFD mice overall travelled less than LFD mice (*p* < 0.05). Dose (F (2, 41) = 3.4, *p* < 0.05) had a significant effect on total distance travelled by female mice, with RK 400 mg/kg mice travelling less than RK 200 mg/kg, see Figure 3C.

### 3.4. Elevated Plus Maze After 14 Days of Dosing and Diet Access

Dose (F (2, 56) = 4.1, *p* < 0.05) had a significant effect on time spent in the open arms by male mice, with RK400 mice spending less time than VEH (*p* < 0.05) in open arms, see Figure 4A. There was a significant effect of dosing (F (2, 56) = 6.5, *p* < 0.01) on number of entries into open arms, with RK400 mice entering the open arms less frequently than VEH (*p* < 0.01), see Figure 4C. There were no significant effects of dosing or diet on elevated plus maze parameters of female mice, Figure 4A–D.

### 3.5. Pre-Pulse Inhibition After 14 Days of Dosing and Diet Access

One day after completing the open field test, the mice underwent testing for the pre-pulse inhibition of an acoustic startle. This was measured in mice after three weeks of their respective daily treatment. Data were analyzed separately for each sex using a two-way ANOVA, with diet and dose as variables. There were no significant effects of diet or dose on parameters of startle response in both sexes, see Figure 5A–B.

### 3.6. Hemodynamics Differences After 14 Days of Dosing and Diet Access

For systolic blood pressure (SBP), in males, there was a dose effect (F (2, 62) = 6.2, *p* < 0.005) with a reduction in SBP with 400 mg/kg (*p* < 0.05). In females, there was a dose effect (F (2, 62) = 5.4, *p* < 0.01) for SBP. In females, the 400 mg/kg also reduced SBP (*p* < 0.05), see Figure 6A. For diastolic blood pressure (DBP), in males there was a dose effect (F (2, 62) = 5.2, *p* < 0.01) with a reduction in DBP with 400 mg/kg (p < 0.05). In females, there was a dose effect (F (2, 62) = 4.0, *p* < 0.05) with a reduction in DBP with 400 mg/kg (*p* < 0.05), see Figure 6B. For mean blood pressure (MBP), in males, there was a dose effect (F (2, 62) = 5.7, *p* < 0.05) with a reduction in 400 mg/kg (*p* < 0.05). Similar effects were observed with dose (F (2, 62) = 4.4 *p* < 0.05) in females with a reduction in MBP (*p* < 0.05), see Figure 6C. For heart rate, in males, there was a dose effect (F (2, 61) = 3.4, *p* < 0.05) with 400 mg/kg only different from 200 mg/kg (*p* < 0.05). In females, there was also a dose effect (F (2, 61) = 14.9, *p* < 0.0005) with 400 mg/kg reducing heart rate (*p* < 0.005), see Figure 6D.

### 3.7. c-Fos Immunopositive Cells of The Caudal Hindbrain in Mice Receiving Vehicle or Raspberry Ketone (400 mg/kg) and Diet Access

In the AP, there were no significant effects for dose, diet, or dose x diet, see Table 1. In the NTS, there was only a significant effect for dose (F (1, 19) = 4.7, *p* < 0.05) with mice receiving RK having a greater number of immunopositive cells (*p* < 0.05), see Table 1. The regions with the highest number of immunopositive cells were the mNTS and iNTS, see Figure 7.

## 4. Discussion

RK is marketed in the United States and other countries as a nutraceutical for appetite control and weight loss [18,27,37]. This study sought to investigate the dose-dependent effects of RK on the meal patterns and hemodynamic alterations associated with an obesogenic diet. Our major finding was a dose-differential response in RK’s effects on weight gain, meal patterns, and hemodynamic parameters.

RK has reported effects on weight gain due to an obesogenic diet in male mice when administered by admixture in the diet [16,17] or, in our previous studies, by oral gavage [18]. In these studies differences in weight gain were observed after three to five weeks of exposure to RK. Therefore, to understand the preceding changes that drive the reduced weight gain, we dosed animals for two weeks with a 200 mg/kg RK and high dose of RK 400 mg/kg. The dose of 400 mg/kg in a high-fat driven obesogenic environment has not been studied before. For weight gain, male mice that received 400 mg/kg RK had a reduced gain in body weight. The difference in body weight was observed on 2–5 days of the daily dosing. In females, there was a reduction in gain in body weight with RK 400 mg/kg compared with RK 200 mg/kg. These body weight differences between males and females are not surprising. Female mice are more resilient to the effect of a high-fat diet [38], and we expect to see greater differences in body weight due to RK in a longer exposure to daily dosing. Previously, we demonstrated that there was no difference in the acute oral dosed RK bioavailability in males compared with females [27]. However, metabolism of phenolic compounds can vary between males and females [39], a possibility that is currently being explored with RK.

Previously, we observed a reduction in weight gain with oral RK 200 mg/kg in males fed a high-fat diet for twice as long (i.e., 28 days) [18]. In that study, meal patterns were measured after 21 days of dosing and were not altered with RK 200 mg/kg [18]. In the present study, we analyzed meal patterns separately for dark and light cycles. We observed a dose effect with 200 mg/kg on nocturnal meal patterns in males. In males, the frequency of meals was reduced, but meal size and duration were increased. The 400-mg/kg dose increased the satiety ratio in males and females. Meal frequency was also reduced in females with 400 mg/kg compared with 200 mg/kg. Taken together, our results suggest that RK doses influence meal patterns, specifically meal frequency. Our study demonstrated an increase in meal number, size and duration associated with high-fat diet, which has been previously reported [23,40]. Similarly, diet-induced obesity susceptible rats consume more meals compared with control fed animals [22]. Therefore, nutraceuticals that mitigate the high-fat diet-induced changes in meal parameters can be useful to normalize aberrant feeding. Notably, the differences in RK-induced changes in body weight and meal patterns were not specific to the obesogenic high-fat diet condition.

Weight changes are often a secondary outcome of anxiogenic and antipsychotic agents [24,25]. As such, we examined whether RK-induced weight loss and meal patterns alterations results in anxiety-like behaviors. The major findings were that RK 400 mg/kg decreased total distance traveled in the open field test. RK 200 mg/kg also increased time spent in the center, which is often noted with anxiolytic agents [41]. However, there were no significant findings in the elevated plus maze with 200 mg/kg. In addition, the reductions in entries into both open and closed arms are suggestive that RK 400 mg/kg reduces locomotor activity and does not promote anxiogenic behaviors. Alterations in pre-pulse inhibition have been noted with antipsychotic medications and amphetamines in rodents [34,42,43]. These compounds are also associated with adverse body weight and meal pattern alterations [44,45]. In this study, RK did not alter pre-pulse inhibition, suggesting the weight reduction was not secondary to sensorimotor gating impairments.

FDA-approved obesity medications and nutraceuticals that reduced body weight have been demonstrated to produce adverse cardiovascular outcomes [46,47,48]. Previous studies in rodents have suggested that RK has a cardioprotective role [19,20]. A 28-day treatment of oral RK (100–200 mg/kg) has been shown to prevent isoproterenol-induced cardiac tissue damage and dyslipidemia in rats [19,20]. That is, RK increased cardiac levels of peroxisome proliferator-activated receptor (PPAR)-alpha, and reduced markers of cell death suggesting a protective role [20]. RK shares structural similarity with synephrine [16], which is known to have an effect on hemodynamics [48,49]. As such, our study demonstrated that RK 400 mg/kg reduced systolic blood pressure (BP), diastolic BP, and mean BP. These effects were demonstrated in males and females, regardless of diet. Future studies will examine the blood parameters, such as pro- and anti-inflammatory cytokines and lipid profiles related to these cardiometabolic parameters. Additionally, the present study demonstrated RK 400 mg/kg increased neural activation, as measured by c-Fos immunohistochemistry, of the nucleus of the solitary (NTS). The NTS receives sensory input from the vagal nerve and has been shown to be involved in meal patterns and cardiovascular control [29,30]. Future experiments will be conducted to elucidate the feeding and cardiovascular mechanisms of action and receptor activation of RK on the NTS and other brain regions.

## 5. Conclusions

Our study demonstrated that RK effectively reduced body weight, altered meal patterns, and reduced cardiovascular outcomes. These differences were differentially observed with dose, but they were not specific to the obesogenic diet. These studies suggest that RK might have limited use to *prevent* weight gain and metabolic signatures associated with high-fat diet.

## Figures and Tables

**Figure 1 nutrients-12-01754-f001:**
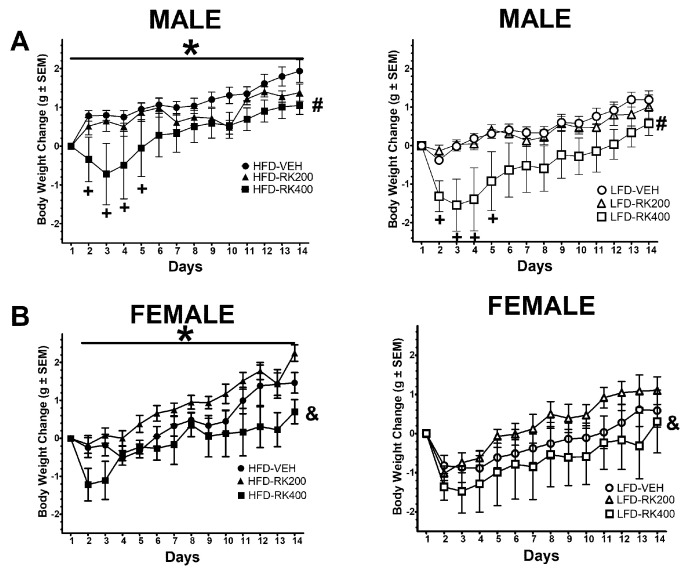
Body weight change in grams over the 14 days of diet access and oral RK dosing compared with baseline. Data are represented as means ± standard error of the mean (SEM). High-fat diet (45% fat; HFD, solid symbols) and low-fat diet (10% fat; LFD, open symbols) and oral gavage with raspberry ketone (RK) or vehicle (50% propylene glycol, 40% water, and 10% dimethyl sulfoxide; DMSO) for 14 days. Comparisons are separate within each sex. (**A**): Males, (**B**): Females. * indicates overall diet difference from LFD (*p* < 0.05), # indicates overall difference from all other doses (*p* < 0.05), + indicates overall daily dose difference from VEH dose (*p* < 0.05), & indicates overall dose difference from 200 mg/kg dose (*p* < 0.05). HFD-Vehicle (males: n = 16, females n = 8), HFD-RK (200 mg/kg) (males: n = 8, females: n = 8), HFD-RK (400 mg/kg) (males: n = 8, females: n = 7), LFD-Vehicle (males: n = 16, females: n = 8) LFD-RK (200 mg/kg)(males: n = 8, females: n = 8), and LFD-RK (400 mg/kg)(males: n = 8, females: n = 8).

**Figure 2 nutrients-12-01754-f002:**
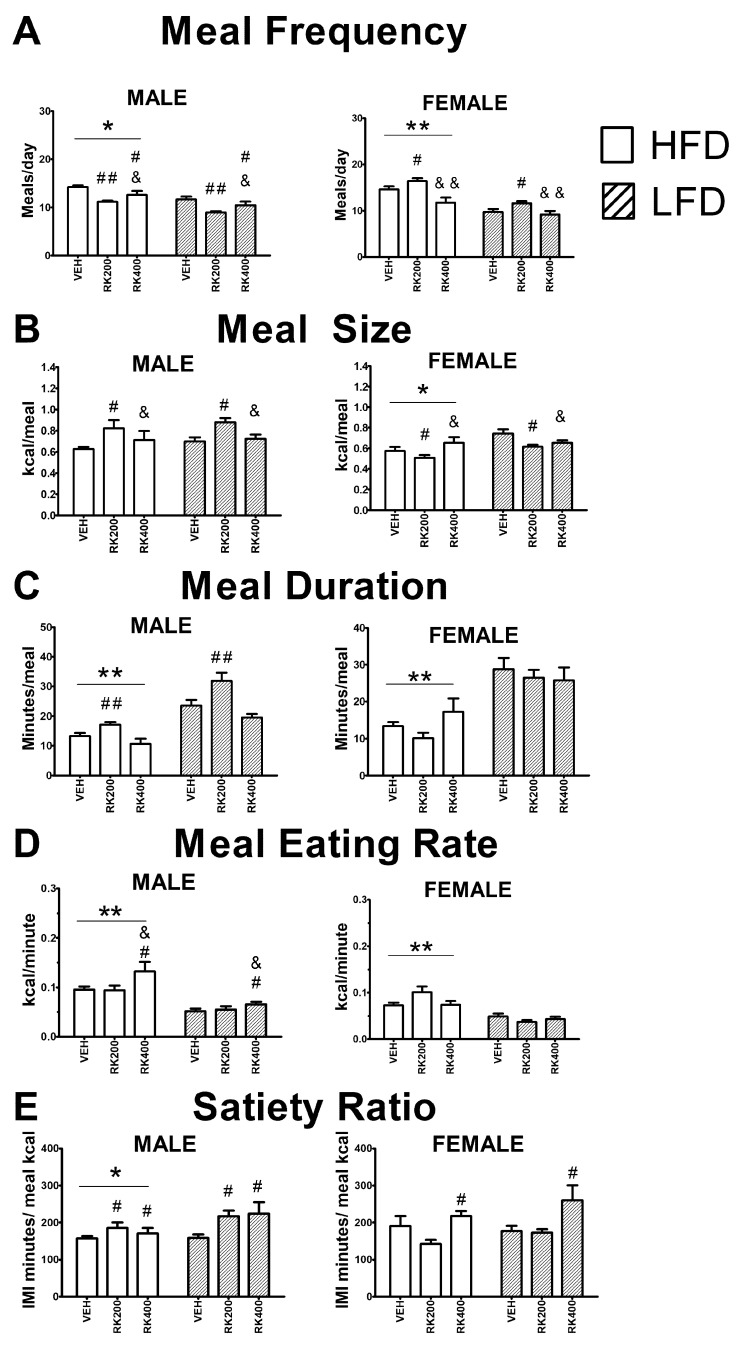
Average meal patterns over the 14 days of daily oral RK and diet access. Meal patterns are from the nocturnal period. Data are represented as means ± standard error of the mean (SEM). High-fat diet (45% fat; HFD, solid lines) and low-fat diet (10% fat; LFD, stripped line) and oral gavage with raspberry ketone (RK) or vehicle (50% propylene glycol, 40% water, and 10% dimethyl sulfoxide; DMSO) for 14 days. Comparisons are separate within each sex. Meal patterns were averaged over fourteen days of dosing and diet exposure. (**A**): Meal frequency; average number of meals each day, (**B**): meal size (kcal), (**C**): meal duration (min), (**D**): meal eating rate (kcal/min), and (**E**): satiety ratio (inter meal interval in min/kcal). * indicates overall diet difference from LFD (*p* < 0.05), ** indicates overall diet difference from LFD (*p* < 0.005), # indicates overall dose difference from vehicle (*p* < 0.05), ## indicates overall dose difference from vehicle (*p* < 0.01), && indicates overall dose difference from RK200 dose (*p* < 0.001), & indicates overall dose difference from RK200 dose (*p* < 0.05). HFD-Vehicle (males: n = 14, females n = 4, HFD-RK (200 mg/kg) (males: n = 8, females: n = 6), HFD-RK (400 mg/kg) (males: n = 7, females: n = 6), LFD-Vehicle (males: n = 16, females: n = 8) LFD-RK (200 mg/kg)(males: n = 7, females: n = 8), and LFD-RK (400 mg/kg)(males: n = 8, females: n = 8).

**Figure 3 nutrients-12-01754-f003:**
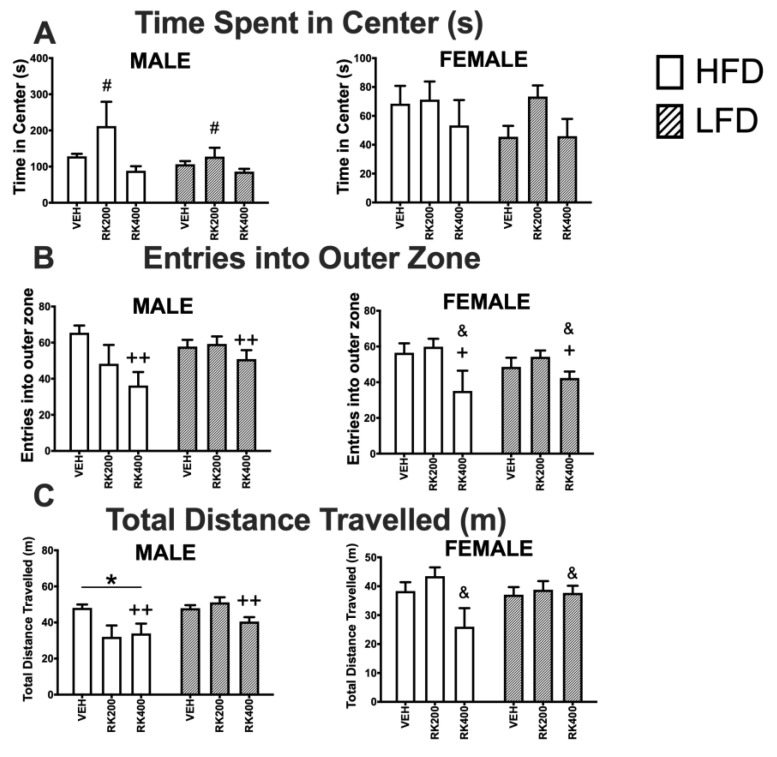
Open-field test during daily oral RK and diet access. Open field tests were performed during days 18–20 of daily dosing and diet access. Data are represented as means ± standard error of the mean (SEM). High-fat diet (45% fat; HFD, solid lines) and low-fat diet (10% fat; LFD, stripped line) and oral gavage with raspberry ketone (RK) or vehicle (50% propylene glycol, 40% water, and 10% dimethyl sulfoxide; DMSO). Comparisons are separate within each sex. (**A**): Time spent in center of open field (s), (**B**): number of entries into outer zone, (**C**): total distance travelled (m). * indicates overall diet difference from LFD (*p* < 0.05), # indicates overall dose difference from all other doses (*p* < 0.05), + indicates overall dose difference from VEH dose (*p* < 0.05), ++ indicates overall dose difference from VEH dose (*p* < 0.01), & indicates overall difference from RK200 dose (*p* < 0.05). HFD-Vehicle (males: n = 15, females n = 8), HFD-RK (200 mg/kg) (males: n = 7, females: n = 8), HFD-RK (400 mg/kg) (males: n = 8, females: n = 7), LFD-Vehicle (males: n = 16, females: n = 8) LFD-RK (200 mg/kg) (males: n = 7, females: n = 8), and LFD-RK (400 mg/kg)(males: n = 8, females: n = 8).

**Figure 4 nutrients-12-01754-f004:**
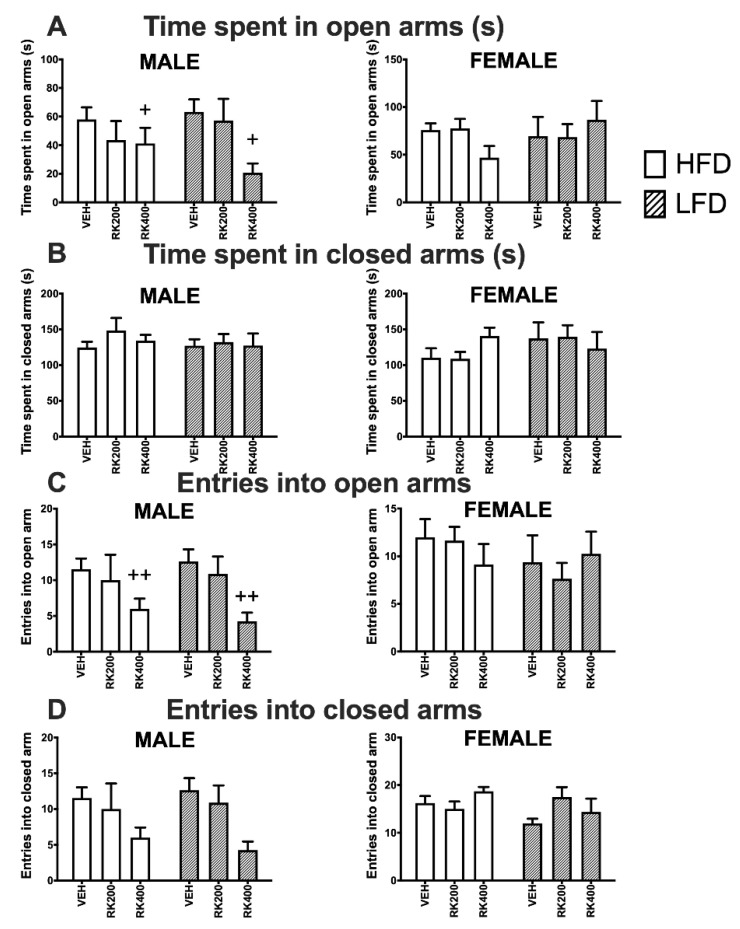
Elevated plus maze behavior test during daily oral RK and diet access. Elevated plus maze tests were performed during days 19–21 of daily dosing and diet access. Data are represented as means ± standard error of the mean (SEM). High-fat diet (45% fat; HFD, solid lines) and low-fat diet (10% fat; LFD, stripped line) and oral dosed with raspberry ketone (RK) or vehicle (50% propylene glycol, 40% water, and 10% dimethyl sulfoxide; DMSO). Comparisons are separate within each sex. (**A**): Time spent in open arms (s), (**B**): time spent in closed arms (s), (**C**): number of entries into open arms, (**D**): number of entries into closed arms. + indicates overall dose difference from VEH dose (*p* < 0.05), ++ indicates overall dose difference from VEH dose (*p* < 0.01). HFD-Vehicle (males: n = 15, females n = 8), HFD-RK (200 mg/kg) (males: n = 7, females: n = 8), HFD-RK (400 mg/kg) (males: n = 8, females: n = 7), LFD-Vehicle (males: n = 16, females: n = 8) LFD-RK (200 mg/kg)(males: n = 8, females: n = 8), and LFD-RK (400 mg/kg)(males: n = 8, females: n = 8).

**Figure 5 nutrients-12-01754-f005:**
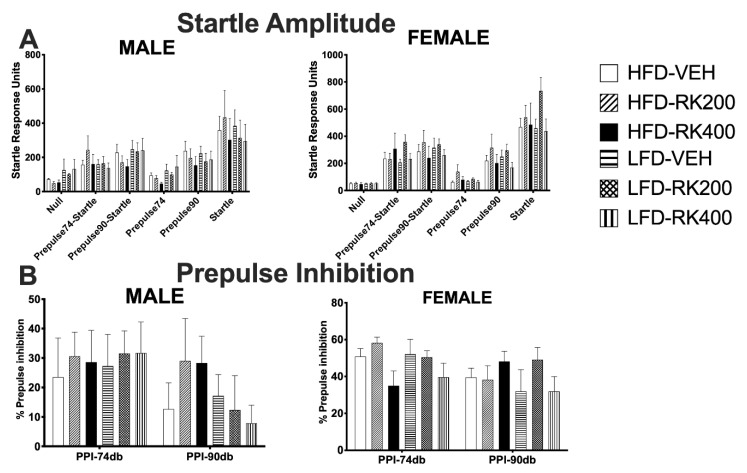
Pre-pulse inhibition of an acoustic startle response behavior test. Pre-pulse inhibition tests were performed during days 21–22 of daily dosing and diet access. Data are represented as means ± standard error of the mean (SEM). High-fat diet (45% fat; HFD, solid lines) and low-fat diet (10% fat; LFD, stripped line) and oral dosed with raspberry ketone (RK) or vehicle (50% propylene glycol, 40% water, and 10% dimethyl sulfoxide; DMSO). Comparisons are separate within each sex. (**A**): Startle amplitude, (**B**)**:** pre-pulse inhibition. HFD-Vehicle (males: n = 15, females n = 8), HFD-RK (200 mg/kg) (males: n = 7, females: n = 8), HFD-RK (400 mg/kg) (males: n = 8, females: n = 7), LFD-Vehicle (males: n = 16, females: n = 8) LFD-RK (200 mg/kg)(males: n = 8, females: n = 8), and LFD-RK (400 mg/kg)(males: n = 8, females: n = 8).

**Figure 6 nutrients-12-01754-f006:**
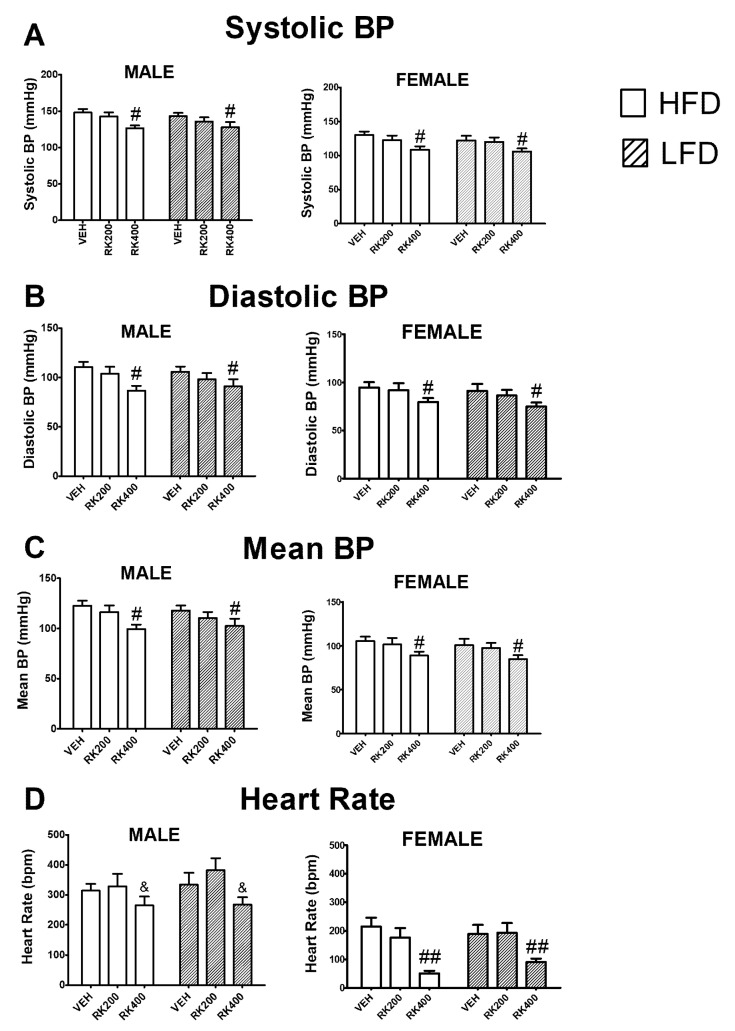
Hemodynamic measurements during the last two days of the 14-day daily oral RK and diet access. Blood pressure and heart rate were measured by noninvasive tail-cuff method. Data are represented as means ± standard error of the mean (SEM). High-fat diet (45% fat; HFD, solid lines) and low-fat diet (10% fat; LFD, stripped line) and oral dosed with raspberry ketone (RK) or vehicle (50% propylene glycol, 40% water, and 10% dimethyl sulfoxide; DMSO) for 14 days. Comparisons are separate within each sex. Hemodynamic measurements were averages over the last 2 days of dosing and diet exposure. (**A**): Systolic blood pressure (mm Hg), (**B**): diastolic blood pressure (mm Hg), (**C**): mean blood pressure (mm Hg), (**D**): heart rate (beats per minute; BPM). # indicates overall dose difference from vehicle and RK 200 mg/kg (*p* < 0.05), ## indicates overall dose difference from vehicle and RK 200 mg/kg (*p* < 0.005) & indicates overall dose difference from RK200 dose (*p* < 0.05). HFD-Vehicle (males: n = 12, females n = 12), HFD-RK (200 mg/kg) (males: n = 12, females: n = 11), HFD-RK (400 mg/kg) (males: n = 11, females: n = 11), LFD-Vehicle (males: n = 11, females: n = 11) LFD-RK (200 mg/kg)(males: n = 11, females: n = 11), and LFD-RK (400 mg/kg) (males: n = 11, females: n = 12).

**Figure 7 nutrients-12-01754-f007:**
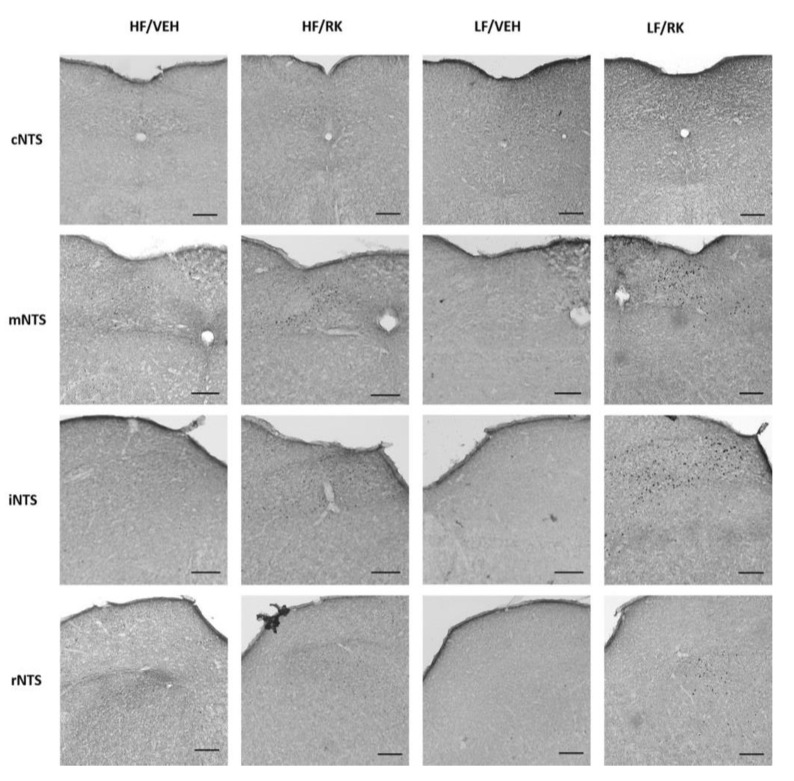
Representative coronal micrographs from NTS and AP in mice dosed with vehicle or raspberry ketone (400 mg/kg). Immunopositive c-Fos cells are stained black. NTS subareas are caudal (cNTS; −7.92 mm from Bregma); medial (mNTS; −7.48 mm from Bregma); intermediate (iNTS; −7.08 mm from Bregma); rostral (rNTS; −6.84 mm from Bregma). AP are located mNTS. Black bar in each image represents 153 μm.

**Table 1 nutrients-12-01754-t001:** Average immunoreactive c-Fos counts in the area postrema (AP) and nucleus of the solitary tract (NTS) from mice orally dosed with vehicle or raspberry ketone (400 mg/kg). Mice were fed high-fat diet (HFD; 45% fat) or low-fat diet (LFD: 10% fat) and orally dosed vehicle (VEH) or raspberry ketone (400 mg/kg) for 14 days. Mice were euthanized on day 14, 120 min after respective dosing. Immunopositive cell counts are means ± SEM.

Brain Region	HFD-VEH (n = 6)	HFD-RK (n = 6)	LFD-VEH (n = 5)	LFD-RK (n = 6)
AP	39 ± 4	30 ± 5	20 ± 12	27 ± 10
NTS	72 ± 30	272 ± 85#	15 ± 28	184 ± 35#

# overall effect of RK (400 mg/kg) compared with vehicle; *p* < 0.05.

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
