# Peer review of "Raspberry Ketone [4-(4-Hydroxyphenyl)-2-Butanone] Differentially Effects Meal Patterns and Cardiovascular Parameters in Mice"

_nutrients, 2020, doi:10.3390/nu12061754_

Round 1
Reviewer 1 Report
Minor revisions
Figure 1A and 1B for HFD are difficult to read. These should be edited
Major revisions
There is a clear difference between female and male response to the RK. This should be investigated more in detail or at least commented in the discussion section.
At 398 line, the authors claimed that the RK reduces systolic and diastolic blood pressure and heart rate. However, as it clear from figure 6, this is not true for all the cases. The author should revise this sentence and discuss more in detail this result. In addition, to connect strongly the RK to the cardiovascular system the authors should consider to perform additional evaluation, such as the analysis of blood biochemical parameters (e.g., pro- and anti-inflammatory cytokines and lipid profiles)
Author Response
Response to Reviewer #1
Figure 1A and 1B for HFD are difficult to read. These should be edited
Authors Response: Yes, we have enlarged Figure 1A and 1B to improve readability.
Major revisions
There is a clear difference between female and male response to the RK. This should be investigated more in detail or at least commented in the discussion section.
Authors Response: Yes, we have added additional text in the Discussion section (lines #396-402).
At 398 line, the authors claimed that the RK reduces systolic and diastolic blood pressure and heart rate. However, as it clear from figure 6, this is not true for all the cases. The author should revise this sentence and discuss more in detail this result.
Authors Response: Thank you. We have corrected this statement since heart rate with RK 400 mg/kg was only different from RK 200 mg/kg in males (line #434-435)
In addition, to connect strongly the RK to the cardiovascular system the authors should consider to perform additional evaluation, such as the analysis of blood biochemical parameters (e.g., pro- and anti-inflammatory cytokines and lipid profiles)
Authors response: Yes, this certainly something we have planned. We have included a statement in the Discussion (line # 436-437).
Reviewer 2 Report
The manuscript by Kshatria et al. assessed the effects of raspberry ketone in high fat diet fed mice on body weight, food intake, behavior and NTS neuronal activation. This paper is the follow up of a similar study that tested the weight lowering effect of a raspberry fruit extract for 28 days.
In this study 2 doses of RK were administrated concomitantly with 45% HFD for 2 weeks in mice. While the results of the study are sound, I would still need clarification on a few majors points:
- Since the RK treatment is sought by overweight people to lose weight, why were the mice not made obese before the treatment ? Indeed the effect on body weight seen during the first week of treatment are similar between LFD and HFD since the 45% HFD fed mice did not have time to gain a large amount of weight compared to LFD ones. Is there a difference in body weight gain between LFD and HFD after 2 weeks ?
- Since the authors mentioned that the RK is rapidly absorbed and peaked 15 min after administration, I am concerned that all the behavioral tests were performed after the treatment stopped. Thus, are the effects seen in the different behavioral tests due to weight loss or to the treatment since they were performed once the treatment has stopped ?
- I suggest the authors to also analyze meal patterns during the first week of treatment since it is when the mice are losing the most weight. Indeed during the second week, it seems that the treatment does not work as well with mice slowly regaining weight.
- The daily food intake data are not presented here but since the experiments were performed in the Biodaq, they should be available. Was there any effect on daily food intake throughout the 2 weeks of treatment ?
- Meal patterns should be separated in light and dark cycle
- Is there a sex difference in the parameters tested ? It would be interesting to know whether males and females respond differently to RK
- Do the authors have any ideas about the receptor for RK and the NTS neuronal subtype activated by RK ?
Author Response
Comments and Suggestions for Authors
The manuscript by Kshatria et al. assessed the effects of raspberry ketone in high fat diet fed mice on body weight, food intake, behavior and NTS neuronal activation. This paper is the follow up of a similar study that tested the weight lowering effect of a raspberry fruit extract for 28 days. In this study 2 doses of RK were administrated concomitantly with 45% HFD for 2 weeks in mice. While the results of the study are sound, I would still need clarification on a few majors points:
- Since the RK treatment is sought by overweight people to lose weight, why were the mice not made obese before the treatment? Indeed the effect on body weight seen during the first week of treatment are similar between LFD and HFD since the 45% HFD fed mice did not have time to gain a large amount of weight compared to LFD ones. Is there a difference in body weight gain between LFD and HFD after 2 weeks?
Authors Response:
This is a very good point. Based on previous research in vitro (Leu et al., 2017; Park, 2010, 2015; Tsai et al., 2017) and in vivo (Cotten et al., 2017; Kshatriya et al., 2019; Morimoto et al., 2005), the effect of raspberry ketone is determined to be preventative as it reduces weight gain in mice or prevents lipid accumulation in 3T3-L1 cells. In our previous research, we have observed significant differences in the prevention of weight gain after about three weeks of dosing (Kshatriya et al, 2019), these experiments were designed to follow-up on those findings. Currently, we are examining whether RK have effectiveness as an obesity treatment, as you described (i.e., making the mice obese first).
To highlight this point, Line 37 now reads: “A few studies reveal the efficacy of RK in the prevention of fat accumulation.”
Yes, there was a difference in weight gain between LFD and HFD, see Figure 1. Post-hoc analysis after a two-way ANOVA revealed that high-fat mice were heavier than low-fat mice at the end of the study (line 201 for males and line 209 for females).
2. Since the authors mentioned that the RK is rapidly absorbed and peaked 15 min after administration, I am concerned that all the behavioral tests were performed after the treatment stopped. Thus, are the effects seen in the different behavioral tests due to weight loss or to the treatment since they were performed once the treatment has stopped?
Authors Response:
We agree, the methods section was not clear about continued daily dosing paradigm through the behavior test days. The doses and diets were continued throughout the behavioral treatments. Open field test, were performed on Days 18-20, elevated plus maze on Days 19-21, and pre-pulse inhibition and Days 20-22. This has been indicated in the figure captions and included in the Results section.
- I suggest the authors to also analyze meal patterns during the first week of treatment since it is when the mice are losing the most weight. Indeed during the second week, it seems that the treatment does not work as well with mice slowly regaining weight.
Authors Response:
Yes, this is a good point. We have reanalyzed the data over first and second week of the treatment. Figure 2 now represents only nocturnal meal patterns data averaged over the two-week study period. Furthermore, to assess the nocturnal feeding of mice and to separate the acute effect of dosing on feeding parameters, we now present nocturnal and diurnal meal pattern data separately.
- The daily food intake data are not presented here but since the experiments were performed in the Biodaq, they should be available. Was there any effect on daily food intake throughout the 2 weeks of treatment?
Authors Response:
Thank you for the suggestion. However, there were no differences in cumulative food intake based on treatment, and the data has been included into the text of Section 3.2 (Line #257-263).
- Meal patterns should be separated in light and dark cycle
Authors Response: Agreed. Meal patterns are now separated for light and dark cycle. Figure 2 represents dark cycle meal patterns. Light cycle meal patterns are described in section 3.2.
- Is there a sex difference in the parameters tested? It would be interesting to know whether males and females respond differently to RK
Authors Response:
We do observe a sex difference for weight gain changes. Female mice from all groups showed a negative weight change which could be attributed to their lower baseline body weight compared with males. This has been discussed in the discussion section (Line# 396-402). However, the meal pattern and hemodynamic parameters in the paper were analyzed separately for males and females. From our previous work on the bioavailability of (Zhao et al., 2019), we do not see a significant difference in bioavailability of RK between the sexes.
- Do the authors have any ideas about the receptor for RK and the NTS neuronal subtype activated by RK ?
Authors Response: Not yet. We are currently working on this in the lab and should have a more definitive answer soon. Anything would be speculative at this point.
We included a sentence on this point (line # 442) at the end of the Discussion.
Round 2
Reviewer 1 Report
It should be interesting whether the authors could add in the discussion some more information addressed to speculate the reason why the metabolism of phenolic compounds can vary between males and females.
If RK metabolism in males and females has not been documented yet in literature, they could comment the metabolism of phenolic compounds in a sex-gender perspective
Reviewer 2 Report
The authors have adequately answered my comments. I have no more comments.